# Spatial chromatin accessibility sequencing resolves high-order spatial interactions of epigenomic markers

Yeming Xie[1†], Fengying Ruan[1†], Yaning Li[1†], Meng Luo[1], Chen Zhang[1], Zhichao Chen[1,2], Zhe Xie[2,3], Zhe Weng[1], Weitian Chen[1,2], Wenfang Chen[1], Yitong Fang[1], Yuxin Sun[1], Mei Guo[1], Juan Wang[1], Shouping Xu[4], Hongqi Wang[1]*, Chong Tang[1]*

[1]BGI Genomics, BGI-Shenzhen, Shenzhen, China; [2]College of Life Sciences, University of Chinese Academy of Sciences, Beijing, China; [3]Department of Biology, Cell Biology and Physiology, University of Copenhagen, Copenhagen, Denmark; [4]Department of Breast Surgery, Harbin Medical University Cancer Hospital, Harbin, China

*For correspondence:
hwang@bgi.com (HW);
leochong718@gmail.com (CT)

†These authors contributed equally to this work

Competing interest: The authors declare that no competing interests exist.

**Abstract** As the genome is organized into a three-dimensional structure in intracellular space, epigenomic information also has a complex spatial arrangement. However, most epigenetic studies describe locations of methylation marks, chromatin accessibility regions, and histone modifications in the horizontal dimension. Proper spatial epigenomic information has rarely been obtained. In this study, we designed spatial chromatin accessibility sequencing (SCA-seq) to resolve the genome conformation by capturing the epigenetic information in single-molecular resolution while simultaneously resolving the genome conformation. Using SCA-seq, we are able to examine the spatial interaction of chromatin accessibility (e.g. enhancer–promoter contacts), CpG island methylation, and spatial insulating functions of the CCCTC-binding factor. We demonstrate that SCA-seq paves the way to explore the mechanism of epigenetic interactions and extends our knowledge in 3D packaging of DNA in the nucleus.

## eLife assessment

This paper reports the development of SCA-seq, a new method derived from PORE-C for simultaneously measuring chromatin accessibility, genome 3D and CpG DNA methylation. Most of the conclusions are supported by **convincing** data. SCA-seq has the potential to become a **useful** tool to the scientific communities to interrogate genome structure-function relationships.

## Introduction

The linear arrangement of DNA sequences usually gives an illusion of a one-dimensional genome. However, the DNA helix is folded hierarchically into several layers of higher-order structures that undergo complex spatial biological regulation. The link between gene transcription activity and genome structure was established following an observation that active gene expression occurs in the decondensed euchromatin, and silenced genes are localized in the condensed heterochromatin (**Klemm et al., 2019**). Accessibility of chromatin acts as a potent gene expression regulatory mechanism by controlling the access of regulatory factors (**Klemm et al., 2019**). Although this model is attractive, it is simplifies the concept of genome accessibility by considering it solely in the horizontal dimension (**Misteli, 2007**). In reality, the genome is organized into a three-dimensional structure within cells, resulting in similar spatial complexities in the accessibility of genome regions. For example, the

accessibility of promoter region could be regulated by the interactions with enhancers or silencers (*Kolovos et al., 2012*). Therefore, it is imperative to employ sophisticated tools that can provide insights into genome accessibility in relation to the organization of the genome, enabling a comprehensive understanding of the interplay between chromatin activation and genome interactions.

Most of the tools designed to study chromatin accessibility in the linear form are based on the vulnerability of open/decondensed chromatin to treatment with enzymes such as DNase, micrococcal nuclease (MNase), and transposase. In a pioneer study, Crawford lab used DNase-seq to establish the relationship between DNase-hypersensitive regions and open chromatin (*Boyle et al., 2008*; *Song and Crawford, 2010*). MNase-seq is based on a similar concept (*Schones et al., 2008*; *Henikoff et al., 2011*). Subsequent studies simplified experiments on chromatin accessibility by taking advantage of the ability of mutant transposase to insert sequencing adapters into open chromatin domains (*Buenrostro et al., 2013*; *Buenrostro et al., 2015*). All these methods rely on statistical calculations of chromatin domain accessibility based on the frequency of enzyme-dependent tags on the accessible genome. Chromatin accessibility has been studied at a single-molecule resolution in recent years to provide insights about chromatin heterogeneity in vivo. Approaches such as methyltransferase treatment followed by single-molecule long-read sequencing (*Wang et al., 2019*), single-molecule adenine methylated oligonucleosome sequencing assay (*Abdulhay et al., 2020*), nanopore sequencing of nucleosome occupancy and methylome (*Lee et al., 2020*), single-molecule long-read accessible chromatin mapping sequencing (*Shipony et al., 2020*; *Chen et al., 2021*), and Fiber-seq *Stergachis et al., 2020* have been developed for this purpose. Generally, decondensed genomes were methylated using methyltransferases and directly sequenced using third-generation sequencing platforms (Nanopore, PacBio). These advanced methods offered a single-molecule view of the two-dimensional 2–15 kb long chromatin structures.

However, it is important to note that chromatin exhibits a higher-order organization, and relying solely on a horizontal dimensional map of chromatin accessibility may not fully capture the complexity of its spatial arrangement. Chromosome conformation capture (3C) techniques, for example, Hi-C, SPRITE, CHIAPET, and pore-C, have been widely used to map genome-wide chromatin architecture (*Bonev et al., 2017*; *Rao et al., 2014*; *Quinodoz et al., 2018*; *Fullwood et al., 2009*; *Deshpande et al., 2022*). Most of these techniques do not allow obtaining epigenome information in the process or examining chromatin conformation with multi-omics. Some advanced approaches, such as Trac-looping (*Lai et al., 2018*), OCEAN-C (*Li et al., 2018*), and HiCAR (*Wei et al., 2020*), can capture open chromatin and chromatin conformation information simultaneously by enrichment of accessible chromatin regions through solubility or transposons. These advanced approaches were originally developed to enrich the accessible chromatin and efficiently observe the interactions of cis-regulatory elements. The loss of the condensed chromatin regions and methylation marks restricted the possibility of observing the full-scale genome architecture with multi-omics. Therefore, reconstructing DNA organization with multi-omics information could promote further understanding of the interactive regulation of transcription and enable more detailed studies of DNA interactions.

Here, we developed a novel tool, spatial chromatin accessibility sequencing (SCA-seq), based on methylation labeling and proximity ligation. The long-range fragments carrying the chromatin accessibility, CpG methylation, and chromatin conformation information were sequenced using nanopore technology. We mapped chromatin accessibility and CpG methylation to genome spatial contacts at single-molecule resolution. Our findings revealed the presence of heterogeneous chromatin accessibility in spatial interactions, suggesting complex genome regulation. We believe that SCA-seq may facilitate multi-omics studies of genome spatial organization.

## Results

### Principle of SCA-seq

Recently, there has been an increasing interest in applying methylation labeling and nanopore sequencing for the analysis of chromatin accessibility at a single-molecule level (*Wang et al., 2019*; *Abdulhay et al., 2020*; *Lee et al., 2020*; *Shipony et al., 2020*; *Chen et al., 2021*; *Weng et al., 2021*). In this study, we have used SCA-seq to study chromatin spatial density to resolve the genome organization and chromatin accessibility simultaneously. (*Figure 1a,1*) After cell fixation, we used a methyltransferase enzyme (EcoGII or M.CviPI) to artificially mark accessible chromatin regions. (*Figure 1a,2-3*

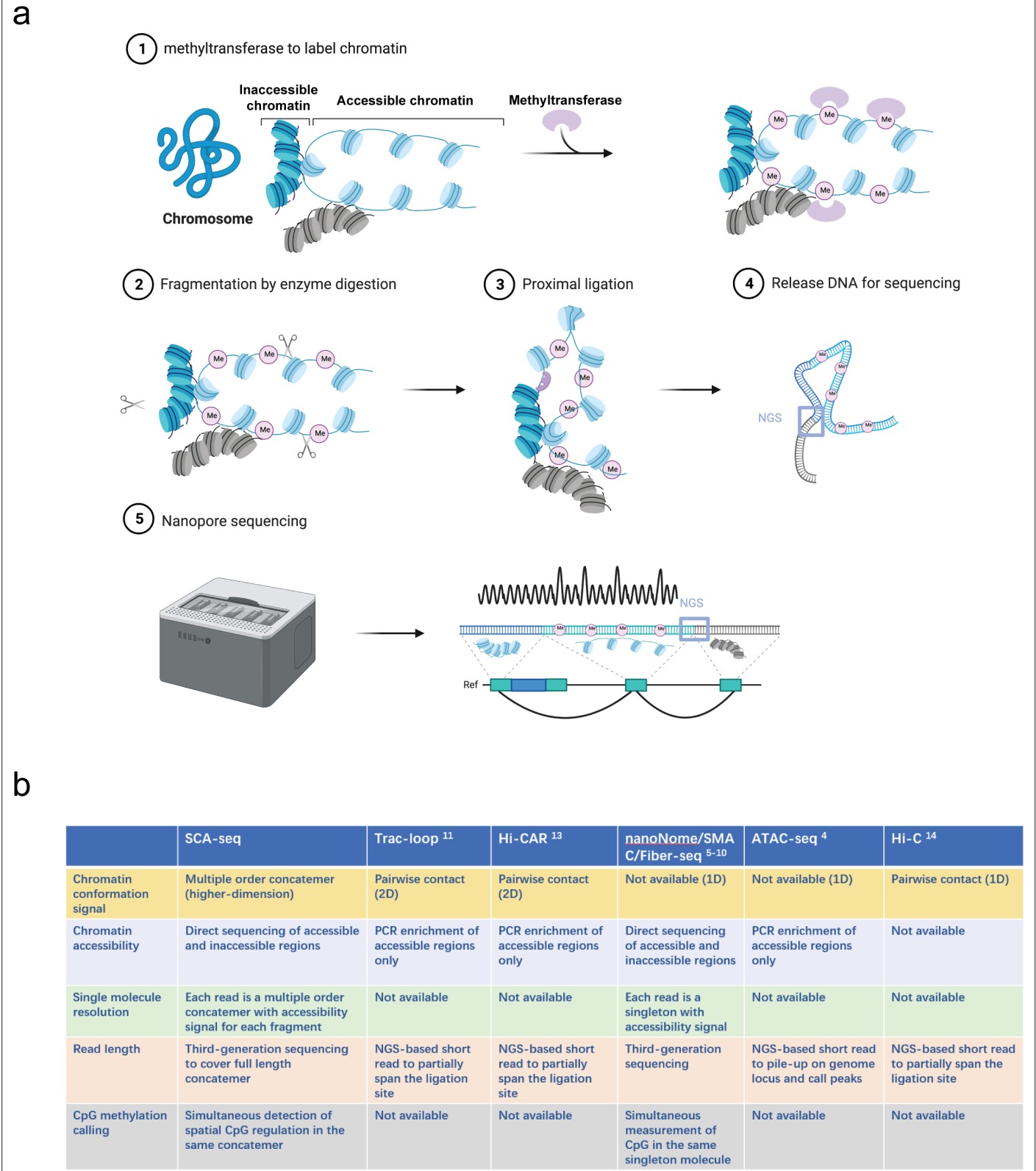

**Figure 1.** Principle of SCA-seq. (**a**) (1) After fixation, chromatin accessibility can be inferred from the extent of artificial methylation by methyltransferase (m6A or GpC, rare in native genomes). (2–4) Restriction enzymes to digest the labeled genome are selected. The interacted segments (>2) stay as a cluster of fragments (concatemer) due to fixation. Next, the spatially interacted segments in one concatemer are proximally ligated, along with the formation of chimeric long fragments. (5) These long chimeric fragments are sequenced using the Nanopore technology. The data-trained Nanopolish

*Figure 1 continued on next page*

*Figure 1 continued*

model helps calling the labeled artificial methylation marks (GpC or m6A), which reflect the regional chromatin accessibility and native CpG on the chimeric reads. Our algorithm analyzes the composition of the chimeric reads, indicating genome locations of these composed segments. Moreover, SCA-seq can simultaneously capture chromatin accessibility and native methylation information of these segments. (**b**) Comparison of advantages of SCA-seq with those of other similar technologies.

The online version of this article includes the following source data and figure supplement(s) for figure 1:

**Figure supplement 1.** Selection of the compatible restriction enzymes.

**Figure supplement 1—source data 1.** Original gel images in *Figure 1—figure supplement 1*.

) After the chromatin accessibility information was preserved as m6A or GpC methylation marks, we conducted digestion and ligation steps using chromatin conformation capturing protocols, relying on proximity ligation to ligate together multiple linearly distant DNA fragments that are close to each other in the three-dimensional space. (*Figure 1a,4*) Next, we performed optimized DNA extraction to obtain pure and large DNA fragments. (*Figure 1a,5*) The DNA fragments that carried chromatin accessibility information, methylation marks, and three-dimensional conformation information were sequenced using the nanopore method and analyzed in our house pipeline from concatemer alignment to single-molecule methylation calling (*Figure 1a*). The conventional next-generation sequencing-based chromatin conformation protocols only captured the interaction between two genomic loci (*Figure 1a*). Unlike the conventional protocol, the proximity ligation in SCA-seq, not limited to the first-order ligation, can occur multiple times in one concatemer (genome fragments fixed together as a cluster), informing about the high cardinality of the genome conformation (*Figure 1a, b*). Compared with the output of the comparable techniques Trac-loop (*Lai et al., 2018*), Hi-CAR (*Wei et al., 2020*), and NicE-C (*Luo et al., 2022*), which also capture the accessible chromatin conformation, the SCA-seq preserved more multi-omics information, for example, CpG methylation epigenetic marks, chromatin inaccessibility, and high-order chromatin conformation data (*Figure 1b*).

First, we experimentally determined the feasibility of SCA-seq. In the methylation reactions, the most suitable methyltransferases, EcoGII and M.CviPI, generated the artificial modifications m6A and m5C (GpC), which are rarely present in the native mammalian genomes (*McClelland and Ivarie, 1982*; *O'Brown et al., 2019*). Our previous research showed that EcoGII effectively labels accessible chromatin owing to the high density of adenine in the genome (*Chen et al., 2021*). However, when EcoGII was used, the high-density labeled m6A modification either blocked or impaired the activity of the majority of restriction enzymes and produced long-digested fragments, leading to lower chromatin conformation resolution (*Figure 1—figure supplement 1a, b*). We also tried digestion with multiple enzymes, as in Hi-C 3.0 (*Lafontaine et al., 2021*), and slightly decreased the fragment length. To solve this problem, we selected the m6A-dependent restriction enzyme DpnI that preferentially digests highly methylated DNA containing methylated adenine and leaves blunt ends. However, such m6A-dependent digestion of the highly methylated accessible chromatin was biased, and the blunt ends were not ligated efficiently. We then tried another approach and used M.CviPI that methylates GpCs (m5C) on the accessible chromatin, and these marks occur four times less frequently than adenosine. In the following steps, DpnII and other enzymes (without GC pattern in the recognition sites) efficiently cut DNA molecules with both methylated and unmethylated GpCs (*Figure 1—figure supplement 1c–e*), followed by ligation. It should be noted that the m5C base-calling algorithm has been gradually improved and is now widely used in nanopore sequencing (*Liu et al., 2021*). Considering the unbiased digestion, M.CviPI might be a better choice in SCA-seq than EcoGII/DpnI. Next, we analyzed the sequencing data and compared them with those obtained using previous technologies.

## SCA-seq has the comparable ability to identify chromatin accessibility and native methylation marks

Our work was inspired by the concepts of nanoNOME-seq, SMAC-seq, and Fiber-seq methods (*Wang et al., 2019*; *Abdulhay et al., 2020*; *Lee et al., 2020*; *Shipony et al., 2020*; *Chen et al., 2021*; *Weng et al., 2021*), which use either M.CviPI or EcoGII methyltransferases to label chromatin-accessible regions with methylation sites. Our previous experiments (*Chen et al., 2021*) and validations of the results against published data confirmed the effectiveness of the methyltransferase-mediated labeling, showing technological advantages of the complex genome alignment and single-molecule resolution

(*Weng et al., 2021*). As the SCA-seq generated discontinuous genomic segments by ligating fragments (*Figure 1*), which might affect data processing, we first validated the accuracy of methylation calling and methyltransferase labeling in SCA-seq.

First, we performed the initial quality control of the sequencing data for HEK293 cells by validating the methylation calling. We generated 129.94 Gb (36.9× coverage) of mapped sequencing data with an N50 read length of 4446 bp. To obtain the methylation information from the nanopore data, we adopted well-established methylation caller Nanopolish with the cpggpc calling module (*Lee et al., 2020*) and achieve considerable success (AUC [Area under the ROC Curve] CpG = 0.908, GpC = 0.984). In the further validation of methylation calling, we parallelly performed the gold standard whole-genome bisulfite sequencing (WGBS). The results of the WGBS analysis were highly correlated with those of Nanopolish (*Figure 2—figure supplement 1*), supporting the accuracy of the methylation caller. In addition to the methylation calling accuracy, there might also be some ambiguity between the native and artificially labeled cytidine methylations. We first checked the native or false positive GpC regions, which were also very rare and accounted for only 1.8% in the unlabeled genome (*Figure 2—figure supplement 2a*). GpC ratio of the M.CviPI-treated genomes was significantly higher than the background GpC ratio (*Figure 2—figure supplement 2b*, >10-fold change, <2.6 × $10^{-16}$, Student's *t*-test). Second, the GCG pattern in the genome might also cause the ambiguity of native methylation CpG or accessibility representing GpC; therefore, we excluded both CpG and GpC methylations from the GCG context (5.6% of GpCs and 24.2% of CpGs) to obtain unbiased methylation information. The excluded GCGs did not significantly affect most of the biological methylation analysis, as reported previously (*Lee et al., 2020*). In conclusion, our bioinformatic pipeline was able to detect native CpG methylations and artificially labeled GpC methylations.

We next assessed the potential of SCA-seq to reveal simultaneously endogenous methylation and chromatin accessibility. As the ATAC-seq and DNase-seq are gold standards for detecting chromatin accessibility, we compared the labeling accuracy of SCA-seq with that of ATAC-seq/DNase-seq globally and locally. Of the 87,991 accessibility peaks called from the SCA-seq data in the whole genome, 80% overlapped with peaks observed in ATAC-seq or DNase-seq (*Figure 2a*). In the evaluation of SCA-seq 1D peak sensitivity and specificity, we considered the non-peak region genomic bins shared by ATAC-seq and DNase-seq as true negatives, and the overlapping peaks of ATAC-seq and DNase-seq as true positives. The obtained results showed a peak sensitivity of 0.73 and a specificity of 0.91 for SCA-seq. These results were comparable to those of a previous study, in which methyltransferase labeling was used (*Lee et al., 2020*). The ATAC-seq/DNase-seq unique peaks were less frequent, as indicated by the larger p-value calculated by MACS2 (*Figure 2b*). On the other hand, the sequencing depth could improve the sensitivity of SCA-seq to identify the less frequent peaks (*Figure 2—figure supplement 3d, e*). Previous publications also suggested that the difference between the outputs of the Nanopore-based and next-generation sequencing-based methods might be explained by sequencing depth (*Lee et al., 2020*; *Shipony et al., 2020*). In the local comparison, SCA-seq also showed peak patterns around the ATAC-seq-identified peaks (*Figure 2c*). Moreover, we computationally predicted binding sites of the CCCTC-binding factor (CTCF), which usually associates with open chromatin (*Ong and Corces, 2014*). As expected, the native CpG methylation level decreased, whereas GpC accessibility increased around the CTCF-binding sites (*Figure 2d*). At active human transcription start sites (TSSs) with high expression, 'open' chromatin regions hypersensitive to transposon attack were observed in ATAC-seq/DNase-seq. SCA-seq showed similar nucleosome depletion patterns around TSSs (*Figure 2e*). Inactive TSSs (low quantile of gene expression) were less accessible than active TSSs (upper quantile of gene expression) (*Figure 2e*). In the detailed examination of the genome regions, the SCA-seq showed the expected nucleosome pattern co-localizing with various epigenetic marks, for example, H3K4me3 (active) and H3K27ac (active) (*Figure 2f* and *Figure 2—figure supplement 4d*). To further estimate the labeling efficiency between different methods, we also compared the fold change values of signal enrichment around TSS with HiCAR, ATAC-seq and DNase-seq, where the accessible chromatin conformation is enriched (*Figure 2—figure supplement 3a*). All three methods SCA-seq, ATAC-seq, and DNase-seq showed similar labeling efficiency on the TSS. We then explored the time-dose influence on the labeling results. The relationship between the dose and M.CviPI treatment effect demonstrated superior efficiency of the 3 hr treatment, comparing with those achieved after 15 or 30 min treatment (*Figure 2—figure supplement 3b, c*). Overall, SCA-seq reliably estimated chromatin accessibility at the genome level.

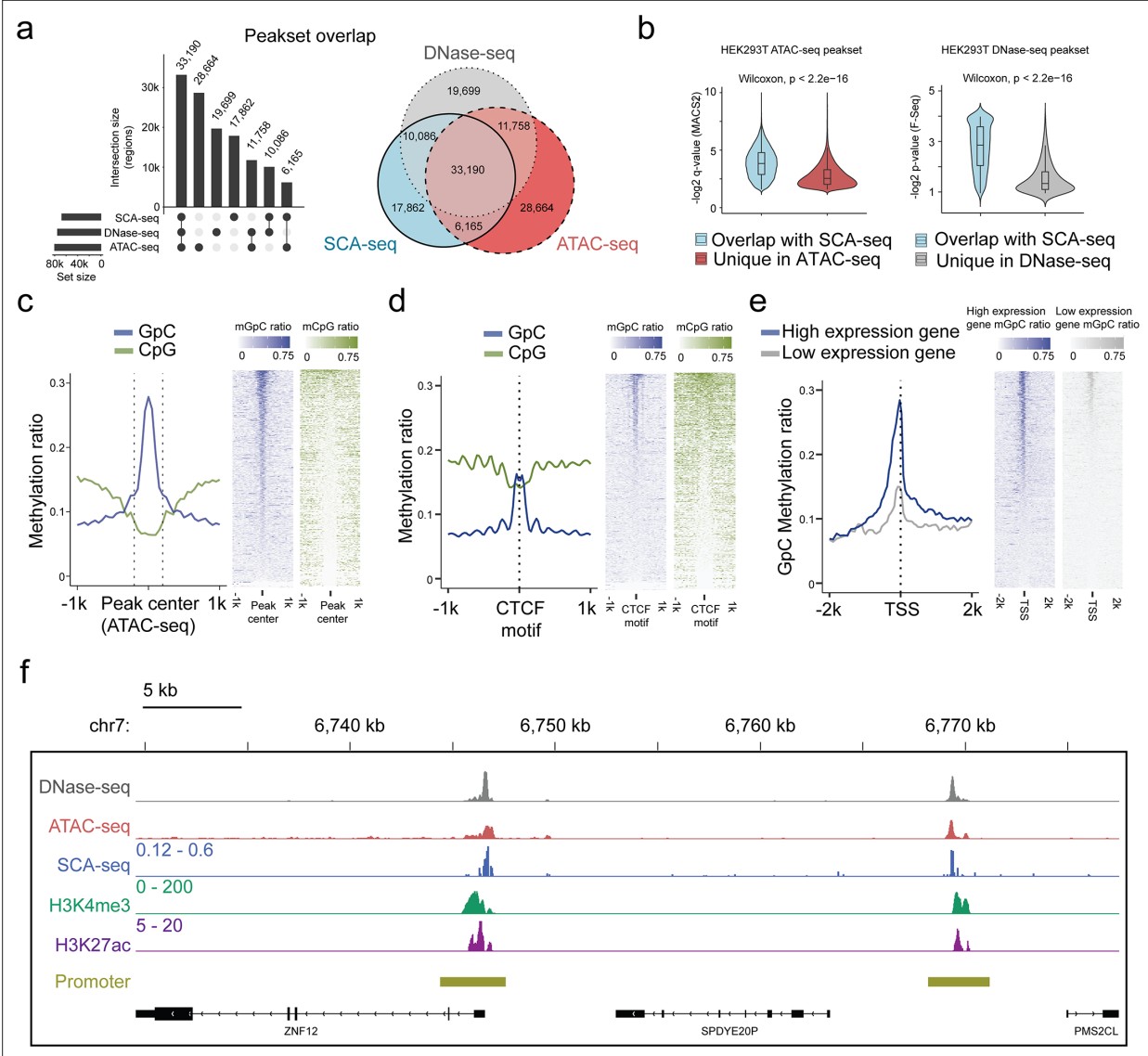

**Figure 2.** Feasibility of accessible chromatin labeling using SCA-seq. (**a**) Venn diagram of the peak overlap between SCA-seq, ATAC-seq, and DNase-seq. The peak-calling algorithms were adapted from nanoNOME-seq. There were 80% peaks identified in SCA-seq overlapped with those detected by either ATAC-seq or DNase-seq. The upset plot is shown on the left. (**b**) Overlapping peaks in the violin plot between SCA-seq and ATAC-seq show the higher MACS2 peak $q$-value than the non-overlapping peaks ($p < 2.6 \times 10^{-16}$, Wilcoxon rank sum test). Similarly, the overlapping peaks in the violin plot between SCA-seq and DNase-seq have the higher F-seq p-value than the non-overlapping peaks ($p < 2.6 \times 10^{-16}$, Wilcoxon rank sum test). (**c**) ATAC-seq peak centered plot. ATAC-seq peak regions were centered at 0 on the $x$-axis and are indicated by the dashed line. The $y$-axis indicates the average methylation ratio on each site (methylation ratio = methylated GpC/all GpC in a 50-bp bin). Blue and green lines indicate the GpC methylation ratio (accessibility) and CpG methylation ratio, respectively. GpC and CpG ratio heatmaps around ATAC-seq peak centers are shown on the right. Each row represents a region of genome with an ATAC-seq peak. (**d**) CTCF motif centered plot at 0 on the $x$-axis indicating the CTCF motif position. The $y$-axis indicates the average methylation ratio among all the molecules across CTCF motifs. SCA-seq demonstrates classic nucleosome depletion patterns around the CTCF motif. GpC and CpG methylation ratio heatmaps around CTCF sites are shown on the right. Each row represents a region of genome with an CTCF site. (**e**) TSS-centered plot. We classified genes with high expression (upper quantile) and low expression (lower quantile) by the expression ranks. GpC and CpG methylation ratio heatmaps around TSSs are shown on the right. Each row represents a region of genome with a TSS. (**f**) A representative genome browser view showing chromatin accessibility signal of DNase-seq, ATAC-seq, SCA-seq, as well as H3K4me3 and H3K27ac ChIP-seq signals in a genome browser (47 kb span). The SCA-seq track shows the GpC methylation ratio in each genomic locus.

The online version of this article includes the following figure supplement(s) for figure 2:

**Figure supplement 1.** Feasibility of methylation calling.

**Figure supplement 2.** GpC labeling efficiency in vitro and in vivo.

**Figure supplement 3.** Dependence of labeling efficiency and signal strength from enzyme concentration and sequencing depth.

*Figure 2 continued on next page*

*Figure 2 continued*

**Figure supplement 4.** Reducing background noise by the detection of accessible regions at segment level.

## SCA-seq reveals high-order chromatin organization

In addition to the methylation information, SCA-seq also preserved genome spatial structure. Therefore, we next validated genome spatial organization. First, we analyzed basic statistical parameters, for example, contact distance and cardinality of SCA-seq. As SCA-seq ligated the multiple fragments together, revealing the multiplex chromatin conformation, we aligned non-singleton chimeric reads into genomic segments and assembled in silico paired-end tags (PETs) in order to compare with Hi-C carrying paired loci. The segment median length was approximately 700 bp (*Figure 3—figure supplement 1a*). Among the informative intra-chromosome PETs, 0.1% of the PETs (contact distance) were <150 bp; 0.3% of them ranged from 150 to 1000 bp; 24.5% were 1000–200,000 bp; and 75.1% were >200,000 bp. Unlike Hi-C, SCA-seq, derived from pore-C, revealed the multiplex nature of chromatin interactions. As for the intra-chromosome interactions, 14.7% of the reads contained two segments (cardinality = 2); approximately 14.5% of the reads contained 3–5 segments (cardinality = 3–5); and 5.4% of the reads had more than five segments (cardinality >5) (*Figure 3—figure supplement 2a*). As in the previous report (*Ulahannan et al., 2019*), most of the contacts from the reads with fewer segments appeared to have closer contact distance (*Figure 3—figure supplement 2b*). The contacts from the reads with more segments appeared to have more distal interactions (*Figure 3—figure supplement 2c, d*). The high cardinality of concatemers with enriched enhancer and/or promoter might indicate the cooperativity in the mammalian transcriptional regulation, as in previous reports (*Ulahannan et al., 2019*). Compared with the output of similar methods, such as Trac-loop and HiCAR, SCA-seq also resolved more high-cardinality chromatin conformation and distal interactions (*Figure 3—figure supplement 2e, f, g*). Our results regarding high-order chromatin conformation capturing also agreed with the output of previously published methods, such as SPRITE (*Quinodoz et al., 2018*), pore-C (*Ulahannan et al., 2019*), and ChIA-Drop (*Zheng et al., 2019*), which also disclosed more distal interactions.

We then compared SCA-seq to the gold standard Hi-C with respect to the false positive call rate, reproducibility, and ability to resolve genome spatial organization. False positive call rates of SCA-seq and Hi-C, inferred from hybrid PETs that consisted of mitochondrial DNA and genomic DNA, were similar (*Figure 3—figure supplement 1b*). The compartment score correlation between SCA-seq replicates and pore-C replicates was approximately 0.94, suggesting comparable reproducibility (*Figure 3—figure supplement 1d*). Furthermore, SCA-seq revealed genome organization similar to the one detected using in situ Hi-C. Side-by-side visualization of interaction heatmaps, loops, topologically associating domain (TAD) boundaries, and A/B compartments obtained using SCA-seq and Hi-C showed equivalent genome organizations (*Figure 3b, c, e, g, h*). The correlation coefficients of the eigenvector and insulation scores were 0.91 and 0.84, that is, slightly lower than we expected (*Figure 3d and f*). From the analysis of a 130G base run of HEK293T SCA-seq, we successfully identified 105,598,180 ligation junctions and 492,502,643 virtual pairwise contacts, which resulted in coverage of 5,625,908 dpnII digested restriction fragments (7,127,633 dpnII in silico restriction fragments). The sequencing depth of SCA-seq could improve these correlations with Hi-C results (*Figure 3—figure supplement 1c*). Because SCA-seq is a non-amplification method, whereas Hi-C is an amplification method, these differences might have resulted from the amplification bias of GC regions (*Figure 3—figure supplement 1e-h*). Notably, we found that 66% of the concatemers were compartment specific (all the segments in one concatemer belonged to either compartment A or B), and 34% were non-specific (mixed composition of segments from compartments A and B) (*Tavares-Cadete et al., 2020*). Overall, these results suggested that SCA-seq successfully resolved the multiplex nature of chromatin interactions.

## SCA-seq resolves the relationship between transcription regulator binding and chromatin conformation

We also could observe chromatin conformation with a specific binding pattern from SCA-seq, for example, the CTCF-binding pattern. As previous publications mentioned, occupation of binding sites by CTCF could lead to the emergence of short regions (~50 bp) inaccessible to methyltransferase

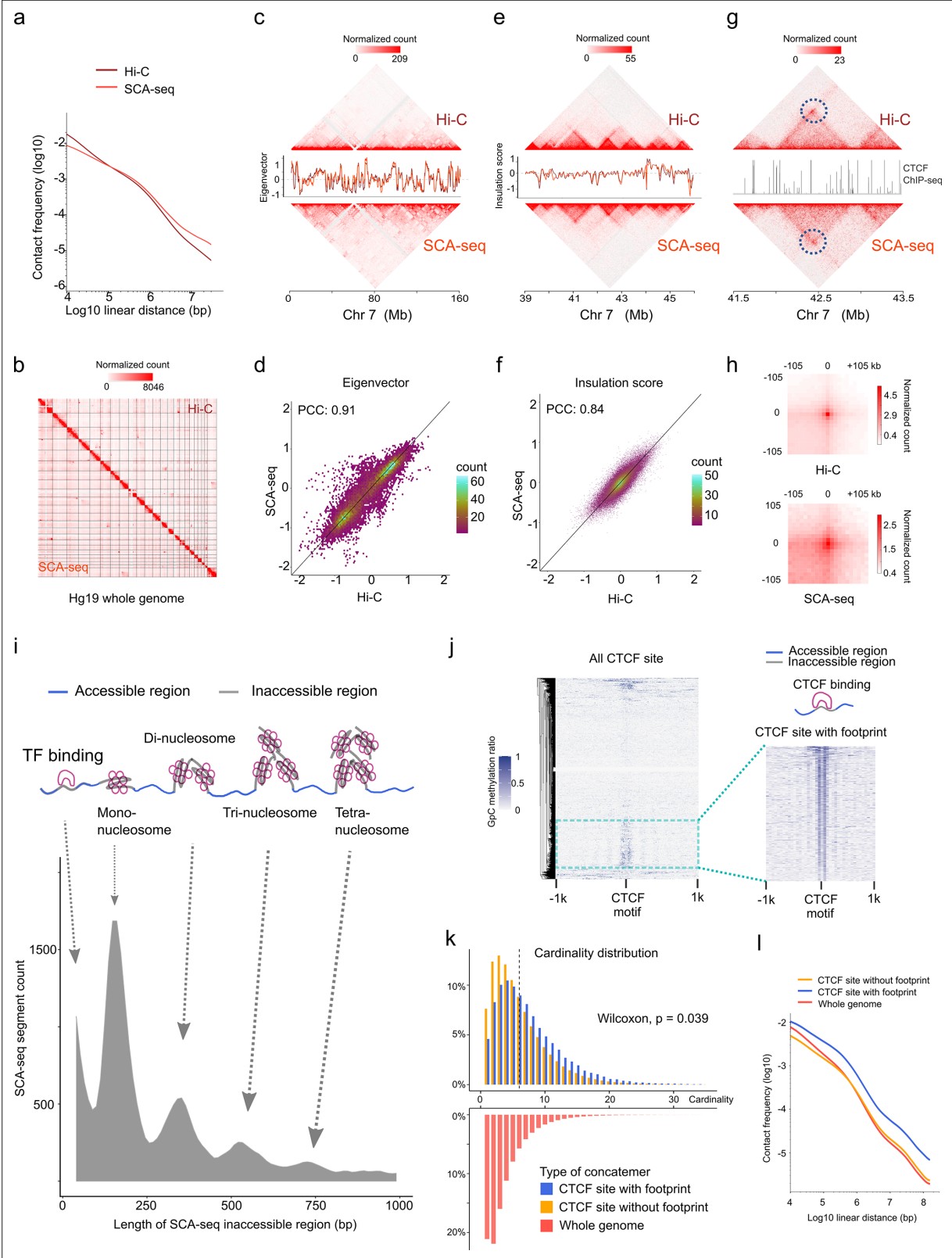

**Figure 3.** SCA-seq captures higher-order chromatin structure and CTCF footprints. (**a**) Contact frequency (*y*-axis) as a function of linear genomic distance (*x*-axis) was plotted across all hg19 chromosomes for SCA-seq (red) and Hi-C (brown). (**b**) Comparison of 1.1 M SCA-seq virtual pairwise contacts (lower triangle) and 1.8 M Hi-C contacts (upper triangle) for chromosomes 1–22 and X (hg19) in HEK293T cells. Comparison of SCA and Hi-C 250 kb (**c**) and 25 kb (**e**) contact maps for chromosome 7. The corresponding eigenvector (**c**) and insulation score (**e**) indicating visual consistency

*Figure 3 continued on next page*

*Figure 3 continued*

between SCA-seq (red) and Hi-C (brown) signal patterns. Color scale bar: log normalized read counts. Scatterplots comparing eigenvector (**d**) and insulation score (**f**) between SCA-seq and Hi-C. PCC: Pearson's correlation coefficient (p < 2.6 × 10⁻¹⁶). (**g**) Contact map showing an example of CTCF peaks at 10 kb resolution with a loop anchor signal indicated by black circles. Color scale bar: log normalized read counts. The CTCF ChIP-seq tract is plotted in between. (**h**) Aggregate peak analysis showing visual correspondence of enrichment patterns between Hi-C and SCA-seq within 100 kb of loop anchors at 10 kb resolution. Color scale bar: sum of contacts detected across the entire loop sets at CTCF sites in a coordinate system centered around each loop anchor. (**i**) The lower density plot shows the inaccessible region length distribution of SCA-seq. The upper panel shows the schematic diagram of the transcription factor binding and nucleosome footprint patterns. (**j**) The heatmap of GpC methylation level shows chromatin accessibility at the CTCF motif at a single-molecule resolution. A subset of CTCF sites with CTCF-binding footprint (50 bp accessible–50 bp inaccessible–50 bp accessible) is shown in a magnified view. (**k**) Cardinality distribution of three sets of concatemers. Concatemers with a CTCF footprint (blue) and without the footprint (yellow) are determined by the CTCF-binding footprint. The genome-wide concatemers are shown in red. Distributions of concatemers with and without CTCF footprint were compared by the Wilcoxon test. (**l**) Multiplex concatemers were converted to pairwise contacts. The linear distances between pairwise contacts are plotted as density distributions.

The online version of this article includes the following figure supplement(s) for figure 3:

**Figure supplement 1.** Quality parameters of SCA-seq for resolving the genome structure.

**Figure supplement 2.** High-order contacts resolve the long-distance genome interaction.

labeling (*Stergachis et al., 2020*; *Battaglia et al., 2022*), indicating the CTCF-binding status on the CTCF motif loci. As expected, SCA-seq also could resolve the transcription factor specific (~50 bp peak) and nucleosome footprints, as was described previously (*Stergachis et al., 2020*; *Battaglia et al., 2022*; *Figure 3i*). Based on the specific accessibility patterns, we classified chromatin interaction concatemers containing CTCF motifs into two classes: with and without a CTCF footprint (*Figure 3j*). Considering the relationship between CTCF binding and chromatin structure formation (*Hyle et al., 2019*), we plotted the concatemer cardinality and interaction distance (*Figure 3k, l*). We found that the CTCF binding resulted in higher cardinality and more distal interactions than the non-CTCF binding, suggesting that CTCF binding facilitates formation of a more complex structure. As has been reported recently (*Battaglia et al., 2022*), the methyltransferase accessibility pattern also could reflect other transcription factors' footprints, enlightening the further exploring the relationship between chromatin conformation with other transcription factors by SCA-seq. Therefore, SCA-seq could help to subgroup chromatin interaction concatemers and investigate the relationship between chromatin conformation and protein binding.

## SCA-seq resolves spatial interactions of accessible and inaccessible chromatin regions

Given the genome organization is highly heterogeneous in different cells, our chromatin interaction status analysis mainly relied on the single-molecule pattern, which needs high sensitivity and specificity. Single-molecule base modification calling was performed as described previously (*Lee et al., 2020*). Moreover, we determined enzyme labeling efficiency, which was 79–88%, based on the CTCF motifs and the lambda DNA spike-in control measurements (*Figure 2—figure supplement 2b, c*). Next, we filtered the segments using the binomial test to minimize false positive attribution of the accessible or inaccessible status (see Methods). Unexpectedly, accessible and inaccessible DNAs were ligated together in SCA-seq (*Figure 4a*), suggesting heterogeneous accessibility of the spatially neighboring DNA regions. Each segment in the concatemer was determined to be either accessible or inaccessible. Then, the fraction of accessible segments in each concatemer was calculated as the $N_{accessible}$/($N_{accessible+inaccessible}$). Compartment A had a significantly higher fraction of accessible segments than compartment B (*Figure 4—figure supplement 1a, b*). Our overall genome concatemer calculations showed that 29% of the genome concatemers were inaccessible on all enclosed segments (the fraction of accessible segments <0.1). Furthermore, 62.2% of genome concatemers had both accessible and inaccessible segments (hybrid concatemers), and only 8.8% maintained all segments as accessible (the fraction of accessible segments >0.9) (*Figure 4b*). *Figure 4a* illustrates an example region with 1D genome feature tracks to demonstrate the promoter–enhancer spatial interactions, accessibility, and CpG methylation at a single-molecule resolution. nanoNOME-seq that labels chromatin accessibility in single molecules also confirmed the existence of partial hybrid concatemers (*Figure 2—figure supplement 2d*). To explore if the concatemer accessibility status was related to spatial location, we plotted the inaccessible concatemers and hybrid concatemers on the 2D contact heatmap. We found

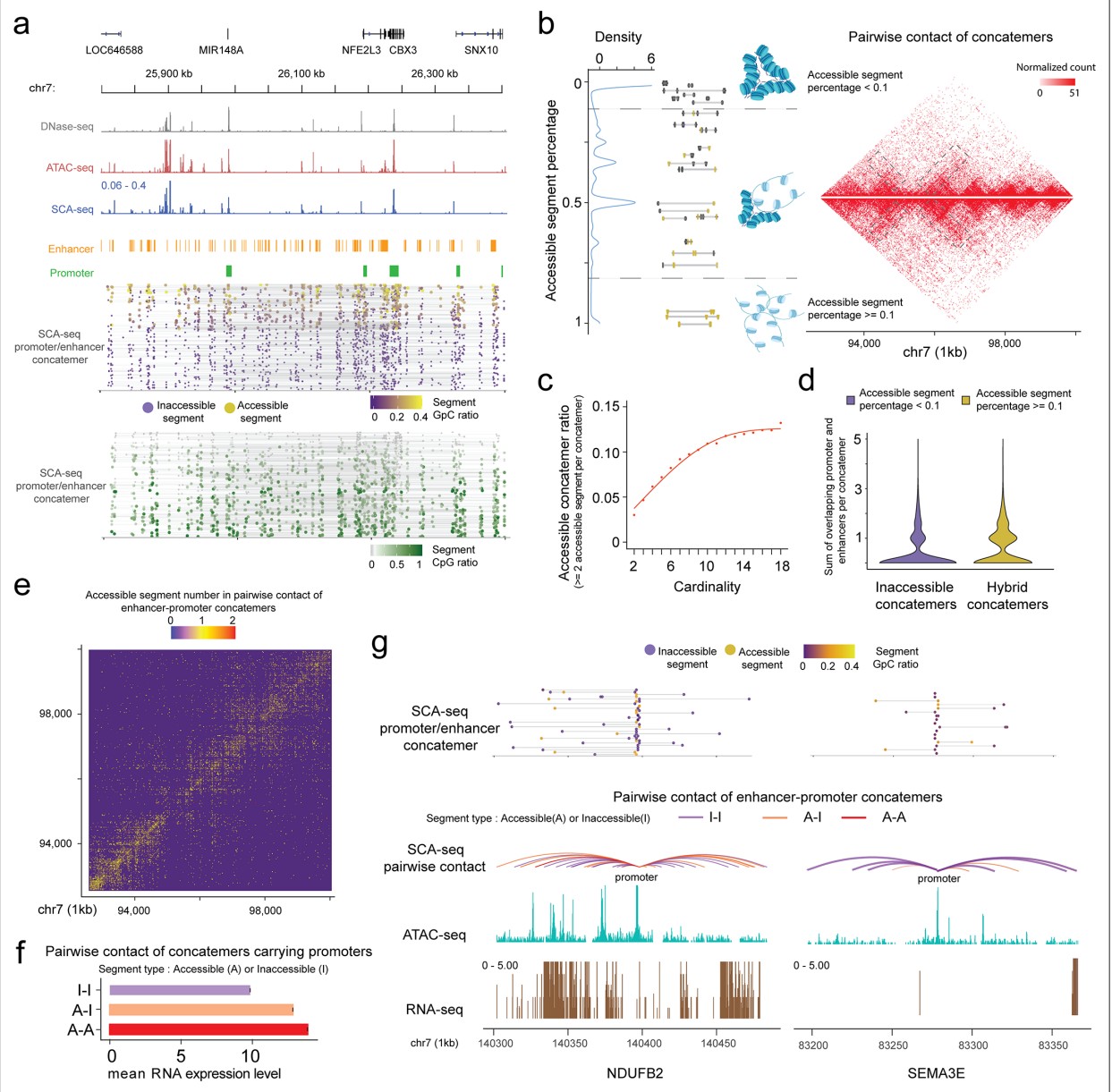

**Figure 4.** SCA-seq demonstrates spatial chromatin accessibility. (**a**) Demonstration of promoter–enhancer concatemers in the region from 25,800 to 26,400 kb on chromosome 7 on a single-molecule level. Gray, red, and blue tracks indicate chromatin accessibility signals from DNase-seq, ATAC-seq, and SCA-seq, respectively. The spatially inaccessible segments that were proximately ligated together to form one concatemer are shown as a single line in the bottom panel. Purple and green dots indicate chromatin accessibility (GpC) and CpG methylation ratio in one segment of a concatemer, respectively. Each row represents an SCA-seq concatemer. (**b**) Distribution of concatemers with specific accessible segment ratios. Each segment was considered accessible if at least one accessible region is identified; otherwise, it was considered inaccessible. Then the ratio of accessible segments was calculated as follows: $N_{accessible}/(N_{accessible+inaccessible})$. The colored line-dot plot shows GpC methylation of segments (dots) in each concatemer (each line). In total, 150,000 pairwise contacts were subsampled for the different types of concatemers in the genome region chr7:92,700,000–100,000,000 with a 25-kb resolution. Inaccessible concatemers tend to accumulate inside the TAD and hybrid concatemers have more TAD boundary interactions and distal interactions (gray box). (**c**) Dot plot with regression curve shows that accessible concatemers (accessible contacts >2 per concatemer) tend to correlate positively with concatemer cardinality (segment number in each concatemer). (**d**) Violin plot of the enhancer/promoter counts in each concatemer. Hybrid concatemers have significantly more enhancers/promoters than inaccessible concatemers (p < 2.6 × 10⁻¹⁶, Wilcoxon rank sum test). (**e**) Accessibility contact map of enhancer–promoter contacts. Inaccessible–inaccessible (I–I), inaccessible–accessible (I–A), and accessible–accessible (A–A) pairwise contacts are shown in blue, yellow, and red colors, respectively. The genome region 92,700,000–100,000,000 is shown at a 25-kb resolution as an example. (**f**) Bar plot showing the relationship of gene expression with different types of accessibility contacts. The y-axis indicates the A–I contact status. The x-axis is the mean RNA expression level the corresponding gene promoters (2 kb upstream of the gene). The accessible enhancer/promoter contacts significantly enhanced gene expression (p < 2.6 × 10⁻¹⁶, Student's t-test). (**g**) An example of two genes with similar accessibility on the promoter

*Figure 4 continued on next page*

*Figure 4 continued*

(2.1% accessibility, promoter-centered plot) but different number of A–A contacts. The curve line track is the chromatin interaction detected by SCA-seq. The cyan and brown tracks are ATAC-seq and RNA-seq by counts on the genome. The top line-dot plot shows the higher-order interaction and accessibility at a single concatemer resolution. The *NDUFB2* gene, with a greater number of accessible contacts, had a notably stronger expression level (528 RPKM) than *SEMA3E* (0 RPKM), which had few accessible contacts.

The online version of this article includes the following figure supplement(s) for figure 4:

**Figure supplement 1.** Correlation between compartmental eigenvalue, RNA expression, and spatial accessibility.

that hybrid concatemers tended to accumulate around the TAD boundaries and contain more distant connections (*Figure 4b*). By plotting the ratio of accessible segments against concatemer cardinality or interaction distance, we revealed that the more accessible segments tended to cluster as high cardinality, also implying their distal and high-cardinality interaction preference on the hybrid concatemers (*Figure 4c* and *Figure 4—figure supplement 1e*). Moreover, we found that the hybrid concatemers comprised more enhancer and promoter elements than inaccessible concatemers (*Figure 4d*), suggesting the relationship between concatemer type and the extent of transcription regulation. We further investigated the enhancer and promoter contacts on chromosome 7, 30.3% of which had accessible–accessible status, 18.5% had accessible–inaccessible status, and 51.2% had inaccessible–inaccessible status (*Figure 4e*). The frequency of contacts with accessible enhancer/promoter was highly correlated with gene expression levels (*Figure 4f, g* and *Figure 4—figure supplement 1c, d*), supporting the transcription model in which active enhancers initiate promoters by spatial contact (*Schoenfelder and Fraser, 2019*). However, 51.2% of enhancer–promoter interactions were independent of the chromatin-accessible status, suggesting that such spatial interactions were not the only factor regulating the initiation of transcription. Overall, the SCA-seq accessible contact signals suggested that spatial interaction and chromatin accessibility might cooperatively regulate gene expression levels.

## SCA-seq resolves CpG methylation on the spatial contacts with orphan CpG islands

CpG islands (CGI), which are widespread features of vertebrate genomes, were associated with ~50% of gene promoters (pCGI). pCGIs control gene transcription by affecting the neighboring promoters with methylation-triggered chromatin changes. Some CGIs are located close to enhancers. In addition, thousands of orphan CGIs (oCGIs), which are at a longer distance (1 kb) from the promoters and enhancers, have been barely known (*Figure 5a*; *Pachano et al., 2021*; *Bell and Vertino, 2017*). In the SCA-seq data that indicated the high-order interaction and CpG methylation, we found 76,418 reads overlapping with CGIs on chromosome 7, and the majority of oCGIs were usually spatially close to the CTCF-binding motifs and active histone markers, such as H3K27ac and H3K4me3, suggesting their active regulatory functions (*Figure 5b*). By examining the methylation status on reads, as expected, these read segments demonstrated lower CpG methylation and higher chromatin accessibility (GpC methylation), which further supports their roles in gene activation (*Figure 5b*). In a previously published study (*Pachano et al., 2021*), oCGIs were considered to act as tethering elements that promote topological interaction between enhancers and distally located genes to regulate gene expression. In SCA-seq, we observed that 60% of oCGIs tethered at least one type of regulatory elements, such as enhancers, CTCFs, and promoters (*Figure 5c*). After normalizing by the total number of regulatory elements, we found that the oCGIs preferentially interacted with CTCF and promoters, comparing with non-CGIs (p < 2.6 × 10$^{-16}$, binomial test, background frequency was used as control) (*Figure 5d*). Further analysis of each concatemer type showed that 39% of oCGI-enhancer concatemers and oCGI-CTCF concatemers included more than two enhancers or CTCF motifs. In contrast, most of the oCGIs tethered to one promoter (*Figure 5e*). However, we found that CpG methylation on oCGIs was weakly correlated with the chromatin accessibility of promoters in regression analysis, whose mechanism of regulation need to be further studied. Overall, oCGIs were found to tether the enhancers and CTCF motifs to communicate with the promoters, which extends our understanding of oCGI regulatory functions.

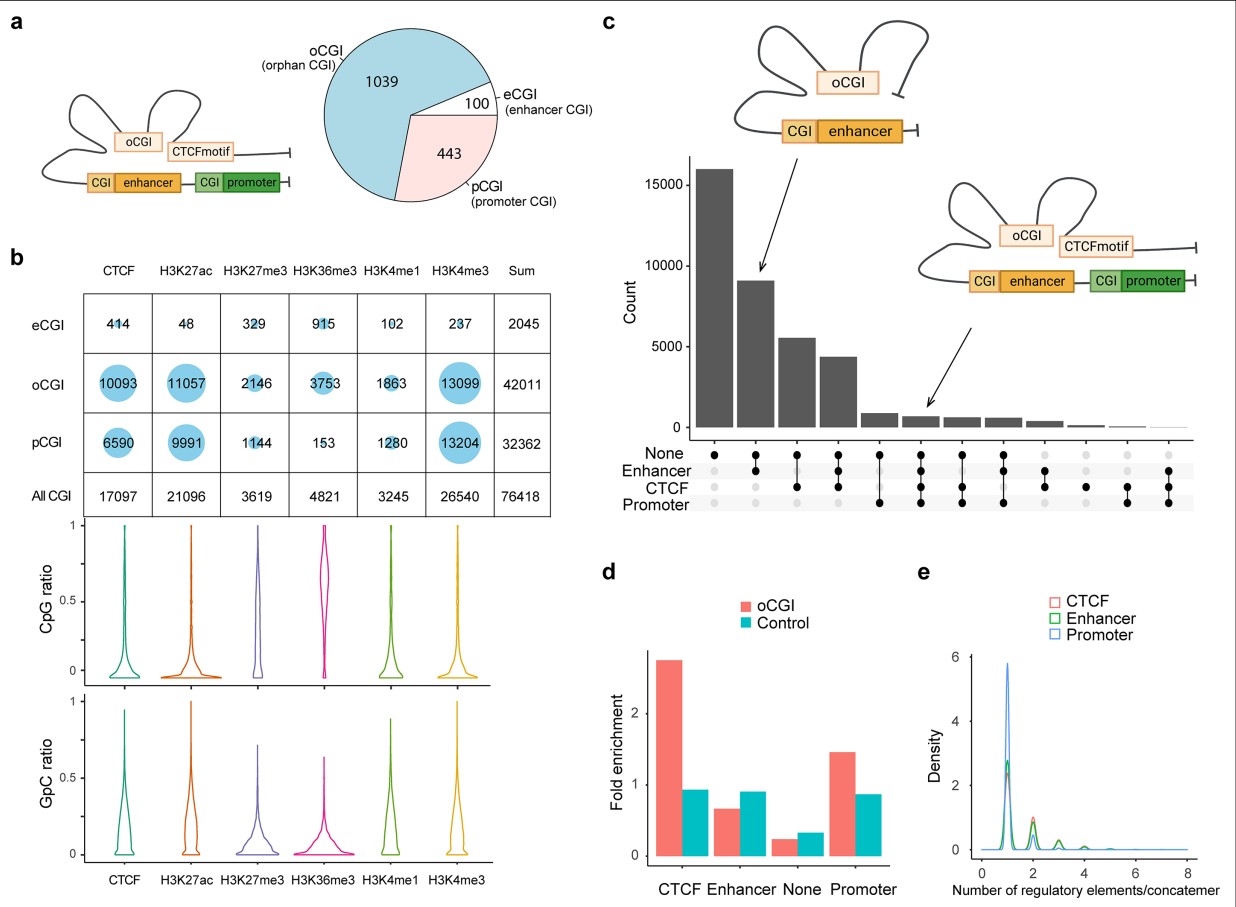

**Figure 5.** CpG methylation and different types of spatial contacts of CpG islands. (**a**) Schematic illustration of orphan CGI (oCGI) spatial contacts with CTCF, promoter, and enhancer. oCGIs are CpG islands that are far away (at least 1 kb) from enhancers or promoters. The enhancer and promoter annotations were downloaded from ENCODE. The eCGIs and pCGIs are defined as CpG islands near or overlapping with the enhancers and promoters. oCGIs comprised 65% of all CGIs that did not directly contact enhancers or promoters. The pie chart shows the abundance of different types of CGIs. (**b**) Number of overlaps between three types of CGIs containing concatemers and histone marks/CTCF site. About 83% of the reads contained oCGI with low CpG methylation level, mainly located with active histone markers that shows high level of chromatin accessibility, such as H3K27ac and H3K4me3, as well as the CTCF-binding motif. H3K27me3 and H3K36me3, which are inactive histone makers, were on the opposite DNA methylation state. (**c**) The upset plots indicate types of interaction between oCGI and other elements in one concatemer. On average, one concatemer has five DNA segments. These segments with oCGIs included CTCF-binding motifs, enhancers, promoters, and other proteins. The *y*-axis indicates the number of concatemers. (**d**) Stacked bar plot shows the abundance of each type of elements interacting with concatemers harboring oCGI or random-sampled concatemers (control). The number of interacting elements was normalized by the total number for itself. oCGIs preferentially interacted with CTCF-binding motifs and promoters. (**e**) Density plot demonstrating the distribution of the number of elements on concatemers. Thirty-nine percent of concatemers have more than one enhancer and CTCF-binding motif with oCGIs. Promoters were in contact only with oCGIs.

## Discussion

We developed SCA-seq to expand the notion of traditional chromatin accessibility to high-dimensional space by simultaneously resolving chromatin accessibility and genome conformation. Compared with 1D ATAC-seq, SCA-seq might more closely represent the true structure of the native genome. With SCA-seq, we found that genome spatial contacts maintained non-uniform chromatin accessibility, suggesting complex genome regulation in the 3D space. SCA-seq could be a multi-omics tool to simultaneously analyzed the DNA methylation, chromatin accessibility, transcription factor binding, and chromatin conformation.

For the single-molecule resolution, the efficiency of methyltransferase labeling must be ensured. We used lambda DNA in vitro labeling and in vivo CTCF motif signal to estimate labeling efficiency. The binomial test was utilized to correct the labeling accuracy at a single-molecule level. Then, relatively reliably accessible chromatin markers were obtained. However, it is still possible for such a

marker to be missed or overridden in the case of insufficient enzyme activity. Because suboptimal labeling efficiency may lead to deviations from our conclusions, our analysis in the following experiments was mainly based on the single-molecule accessible signals from called accessible region to reduce background noise. Given the high heterogeneity of the dynamic genome structure and SCA-seq resolution, a much higher sequencing throughput is required to achieve analysis at a single-molecule level in a specific spatial location. The cost of sequencing a 40× depth human sample using SCA-seq is approximately 1300 USD per sample. The cost may pose a limitation when attempting to perform high-depth SCA-seq profiling of large number of samples. It is important to note that SCA-seq requires the use of a regular Oxford nanopore sequencer with an R9.4.1 chip. While this chip is currently available, there is a possibility that Oxford Nanopore may discontinue it in the future.

The eigenvalue and insulation score of SCA-seq moderately correlated with those of gold standard Hi-C (0.91 and 0.84), and the moderate correlations were also true in all other multiplex-order chromatin conformation methods, such as pore-C (*Deshpande et al., 2022*), SPRITE (*Quinodoz et al., 2018*), and ChIA-Drop (*Zheng et al., 2019*). After we studied this problem deeper, we found that there may be two reasons for this. First, the conversion from multi-contacts to pairwise contacts overrepresented some interactions. In our algorithm, we significantly resolved this issue by weighted transformation, which increased the correlation coefficient to 0.9. Second, we found that the low correlation regions had lower GC density and low read counts (*Figure 2—figure supplement 4e-h*). It has been pointed out previously (*Niu et al., 2019*) that PCR (polymerase chain reaction) amplification could bias eigenvalues and insulation scores. In contrast, pore-C and SCA-seq are non-amplification methods. After the quality filter of low-coverage regions, the correlation between the two methods was significantly improved.

SCA-seq was created as a multi-omics tool to examine both chromosome conformation and chromatin accessibility. It is important to discuss different levels of resolution of chromosome conformation capture and chromatin accessibility. The resolution of the chromosome conformation capture is approximately 700 bp, whereas that of the conventional chromatin accessibility is approximately 200 bp. Not all precise accessible–accessible chromatin interactions can be determined. Therefore, an alternative hypothesis is that the interaction loci are located outside the accessible chromatin. Therefore, improvement of the resolution of chromosome conformation capture in SCA-seq is needed to determine spatial accessibility interaction accurately.

Overall, our results demonstrated that SCA-seq could resolve genome accessibility locations in the three-dimensional space, facilitating the observation of subgroups of chromatin conformation regions with a specific binding pattern, conformation-based chromatin accessibility, and conformation-based native CpG methylation. SCA-seq might pave the way to explore dynamic genome structures in greater detail.

## Methods

The detailed protocol could be found https://www.protocols.io/view/sca-seq-b6a6rahe. The bioinformatic script could be found https://github.com/genometube/SCA-seq (*genometube, 2024*). The data source and QC information could be found in *Supplementary files 1 and 2*.

### Cell culture

Mouse mammary gland carcinoma cell line 4T1 and the derivative human embryo kidney cell line which expresses a mutant version of the SV40 large T antigen (HEK293T) were obtained from ATCC. 4T1 were grown in RPMI1640 (Gibco, 11875093) supplemented with 10% fetal bovine serum (FBS; Gibco, 10099141), and 1% penicillin–streptomycin (Gibco, 15140122). HEK 293T was maintained in Dulbecco's modified Eagle medium-high glucose (Thermo Fisher 11995065) supplemented with 10% FBS (Thermo Fisher 1009141). Cell lines were regularly checked for mycoplasma infection (Yeasen, 40612ES25) and confirmed negative.

### Cross-linking

Five million cells were washed one time in chilled 1× phosphate-buffered saline (PBS) in a 15-ml centrifuge tube, pelleted by centrifugation at 500 × *g* for 3 min at 4°C. Cells were resuspended by gently pipetting in 5 ml 1× PBS with formaladehyde (1% final concentration). Incubating cells at room

temperature for 10 min, add 265 µl of 2.5 M glycine (125 mM final concentration) and incubate at room temperature for 5 min to quench the cross-linking. Centrifuge the mix at 500 × *g* for 3 min at 4°C. Wash cells two times with chilled 1× PBS.

## Nuclei isolation and methylation
Cell pellet was resuspended with cold lysis buffer: 10 mM HEPES (4-(2-hydroxyethyl)-1-piperazin eethanesulfonic acid) –NaOH pH 7.5, 10 mM NaCl, 3 mM MgCl$_2$, 1× proteinase inhibitor (Sigma 11873580001), 0.1% Tween-20, 0.1 mg/ml bovine serum albumin (BSA), 0.1 mM EDTA (Ethylenedi-aminetetraacetic acid), 0.5% CA-630, incubate on ice for 5 min. Centrifuge lysis mixture at 500 × *g* for 5 min at 4°C to collect the nuclei. Washed the nuclei once with 1× GC buffer (NEB M0227L) then resuspend 2 million nuclei in 500 µl methylation reaction mixture: 1× GC buffer, 200 U M.CvipI (NEB M0227L), 96 µM *S*-adenosylmethionine, 300 mM sucrose, 0.1 mg BSA, 1× proteinase inhibitor, 0.1% Tween-20. Incubate the reaction for 3 hr at 37, add 96 µM SAM (S-Adenosylmethionine) and 20 U M.CvipI per hour. Centrifuge at 500 × *g* for 10 min at 4°C to collect nuclei, wash the nuclei once with chilled HEPES–NaOH pH 7.5 and centrifuge to collect nuclei.

## Restriction enzyme digest
Resuspend nuclei with 81 µl cold HEPES–NaOH pH 7.5, add 9 µl 1% sodium dodecyl sulfate (SDS) and react at 65°C for 10 min to denature the chromatin, take the tube on ice immediately after the reaction. We also tried 0.5% SDS, and the results were similar (data not shown). Add 5 µl 20% Triton X-100 and incubate on ice for 10 min to quench SDS. Prepare digestion mixture: 140 U DpnII (NEB R0543L), 14 µl 10× HEPES-buffer3.1 (50 mM HEPES–NaOH pH 8.0, 100 mM NaCl, 10 mM MgCl$_2$, 100 µg/ml BSA), add nuclei suspension and nuclease-free water into mixture to achieve a final volume of 140 µl. Incubate digest mixture in a thermomixer at 37°C for 18 hr with 900 rpm rotation.

## Ligation
DpnII digests were heat inactivated at 65°C for 20 min with 700 rpm rotation, average digests to 70 µl per tube, add 14 µl T4 DNA Ligase buffer (NEB M0202L), 14 µl T4 DNA Ligase (NEB M0202L), 1 mM ATP and nuclease-free water to achieve a final volume of 140 µl. The ligation was incubated at 16°C for 10 hr with 800 rpm rotation.

## Reverse cross-linking and DNA purification
Collect all ligation into one 1.5 ml tube, add equal volume of 2× sera-lysis (2% polyvinylpyrrolidone 40, 2% sodium metabisulfite, 1.0 M sodium chloride, 0.2 M Tris–HCl pH 8.0, 0.1 M EDTA, 2.5% SDS), add 5 µl RNaseA (QIAGEN 19101), incubate at 56°C for 30 min. Add 10 µl Proteinase K (QIAGEN 19131), 50°C overnight incubation with 900 rpm rotation. DNA was purified with high molecular weight gDNA extraction protocol (Baptiste Mayjonade, 2016).

## Nanopore sequencing
The DNA from the reactions were purified, and library were prepared following the manufacturer's protocol of SQK-LSK109 (Nanopore, SQK-LSK109). The library was sequenced in the ONT Prome-thION platform with R9.4.1 flow cell.

## WGBS
The WGBS was parallelly performed from the purified DNA product by using MGIEasy Whole Genome Bisulfite Sequencing Library Prep Kit (MGI 1000005251). The final products were sequenced by MGISEQ-2000.

## SCA-seq pipeline
We developed a reproducible bioinformatics pipeline to analyze the M.CvipI footprint and CpG signal on SCA-seq concatemers. Briefly, the workflow starts with the alignment of SCA-seq reads to a reference genome by bwa (v0.7.12) using the parameter bwa bwasw -b 5 -q 2 -r 1 -T 15 -z 10. The mapping score ≥30, and reads with length <50 bp were set to filter out the low-quality mapping segment. To remove the non-chimeric pairs due to ligation of cognate free ends or incomplete digestion, each alignment is assigned to an in silico restriction digest based on the midpoint of alignment. The locus of

each segment on each concatemer is summarized by converting the filtered alignment to a segment bed file sorted by read ID first and then the genome locus. The alignment bam file is also used to call the GpC and CpG methylation by Nanopolish (v0.11.1) call-methylation with the cpggpc model (--methylation cpggpc). The default cut-off for log-likelihood ratios are used to determine methylated GpC (>1) and methylated CpG sites (>1.5) (*Lee et al., 2020*). The methylation call is then counted to each segment in the segment bed file to derive the methylated and unmethylated count of GpC and CpG for each segment of the concatemers.

To filter out background GpC methylation signal, we performed a segment-level sliding window analysis combined with a binomial test. The GpC methylation signal was divided into 50 bp sliding windows with a 10-bp step size. For a given window, let the background GpC methylation ratio be the probability of success in a given trail, where the number of methylated GpC sites be the number of successes and the total number of GpC sites be the number of trials. A right-tailed binomial test was performed to determine if the GpC methylation ratio in the given window is significantly greater than the background GpC methylation ratio ($p < 0.05$). The accessible region on the given segment was then defined by merging the overlapping windows that pass the binomial as well as contain at least 2 GpC sites. Once the accessible regions on each segment of each read is determined, the GpC methylation signal out of the accessible regions are removed to reduce the random noise in GpC methylation signal. For the CTCF footprint analysis, we identify CTCF site with footprints on each segment if the accessible regions are called nearby the CTCF sites (at least 20 bp away from the center of CTCF sites) but not on the CTCF sites.

## SCA-seq and Hi-C comparisons

SCA-seq concatemers were converted into virtual pairwise contacts in order to correlate with the published Hi-C datasets. For a given pair of two segments in an SCA-seq concatemer, the number of segments between the two segments ($k$) is positively correlated with the spatial distance between them and negatively correlated with the Hi-C observable counts. Therefore, we down-weighted each of the two segments in an SCA-seq concatemer such that each pair of two segments has a weight of $1/2^k$. The decomposed SCA-seq contact matrix was treated as a Hi-C contact matrix and analyzed by Cooltools (v0.4.1) (*Abdennur, 2022*). The contact matrix was normalized using cooler balance. Then the eigenvector scores and TAD insulation score were calculated by Cooltools call-compartments and Cooltools diamond-insulation tools. The linear correlation between the pore-C and Hi-C contact matrices was then measured by eigenvector scores (compartment score) and TAD insulation score calculated by Cooltools. The variation of individual pore-C runs, individual SCA-seq runs, and downsampled SCA-seq datasets were also examined by the above metrics. Loop anchors were identified by ENCODE CTCF ChIP-seq peaks (ENCSR135CRI). Cooltools pile-up was used to compute aggregate contact maps at 10 kb resolution and centered at the loop anchors (±100 kb).

## SCA-seq, ATAC-seq, and DNase-seq comparisons

For comparison and visualization of bulk accessibility, the conventional bulk ATAC-seq and DNase-seq data of HEK293T peak signals were obtained from Gene Expression Omnibus (GEO) accession GSE108513 and GSM1008573. The SCA-seq accessibility peak calling was performed in a similar way to nanoNOMe (*Lee et al., 2020*). Briefly, 200 bp window and 20 bp step size continuous regions of GpC methylated counts, unmethylated counts, and GpC methylation ratio were generated from SCA-seq Nanopolish calls. The regions of GpC methylation ratio greater than 99th percentile of the regions were selected as candidate first. The significance of each candidate region was calculated by the one-tailed binomial test of raw frequency of accessibility (methylated GpC site/total GpC site) to reject the null probability, which is defined by the overall regions GpC methylation ratio. The p-values were corrected for multiple testing by Benjamini–Hochberg correction. The adjusted p-values <0.001 and widths greater than 50 bps were determined as the SCA-seq accessibility peaks. The overlapping peaks between SCA-seq, ATAC-seq, and DNase-seq were identified by bedtools (v2.26.0) intersect. The CTCF-binding sites, enhancer regions, and promoter regions are obtained from the Ensembl Regulatory Build (*Zerbino et al., 2015*) in the 1D track view and pile-up analysis.

## Analysis of cytosine modifications called by WGBS

The cytosine modification analysis of WGBS data generally follows the rules in *Heyn et al., 2012*. In brief, the two sets of hg19 genome reference sequences were prepared; the C to T reference that had the C residues replaced by Ts, and the G to A reference that had the T residues replaced by As. The two sets of reads are prepared the same way; the C to T reads and G to A reads. The two sets of reads were aligned to the two sets of genome separately by GEM alignment software (*Marco-Sola et al., 2012*) which allows 4 mismatches of bases with quality score over 25. The lack of methylation of C residues would be recognized as C to T or G to A conversions. The methylation ratio of methylated GpC or CpG residues in 200 bp windows was then calculated for the correlation analysis with the SCA-seq methylation calling.

## Estimate the labeling efficiency in vivo

As previous research, the CTCF motif maintained the accessible chromatin in neighboring 200 bp region. Consider the resolution in Hi-C and experimental fragmentation, we selected the 1000 bp bins with the documented CTCF motif in center. The CpG methylation levels were negatively correlated with the chromatin accessibility. Then the segment with low CpG methylation were expected to maintain the accessible chromatin status with CTCF binding. We hypothesized that the segments with low CpG methylation (CpG ratio <0.25) and low chromatin accessibility (GpC ratio <0.1) were not efficiently labeled.

## Identification of accessible segments by binomial test

The medium segment length is 500 bp, which is close to the general size of accessible chromatin segments. We first calculated the background level of the methyl-GpC (accessible) and non-methyl-GpC (inaccessible) probability on the segments. We used the non-treated genomic DNAs as the background, and 0.03 (GpC background) were the average GpC frequency on segments. Then we performed the binomial test (R basics) for each segment in M.CviPI-treated samples to test the null hypothesis that if labeled GpCs (GpC ≥4) was equal or smaller than the background GpCs. We further to investigate the confidence level of inaccessible chromatin with the non-methyl GpC. The non-methyl-GpC frequency in M.CviPI-treated spike-in is 0.3. Therefore, we roughly estimated that 21% GpCs(p) were not efficiently labeled by M.CviPI. Then we performed the binomial test (R basics) for each segment in M.CviPI-treated samples to test the null hypothesis that if the non-methyl GpCs on heterochromatins were equal or larger than the enzymatic inefficiency. For both p-value, the probabilities were corrected for multiple testing using the Benjamini–Hochberg correction and accessible/inaccessible segments with adjusted p-value less than 0.05. We determined the accessible segments first, and then we further determined the inaccessible segments in the rest. There are ~2 million segments which is undetermined and discarded.

## Statistics

Most of the parametric data which were distributed as normal distribution (log normal distribution), were performed in two-side *t*-test. The Pearson's correlation analysis was also performed for normal distribution data. We used the Fisher's exact test for the differential accessibility analysis in SCA-seq. Other non-parametric or abnormally distributed data were performed as Wilcoxon rank test.

## Data availability

The data were stored at https://db.cngb.org/search/project/CNP0002862/ and NCBI BioProject PRJNA917827. All custom codes for SCA-seq are available from GitHub https://github.com/genometube/SCA-seq.

## Acknowledgements

This research was supported by the Science, Technology, and Innovation Commission of Shenzhen Municipality (grant number JSGG20170824152728492). The supporter had no role in designing the study, data collection, analysis, and interpretation, or in writing the manuscript.

## Additional information

### Funding

| Funder | Grant reference number | Author |
| --- | --- | --- |
| Shenzhen Municipal Science and Technology Innovation Council | Grant number | Mei Guo |
| Shenzhen Municipal Science and Technology Innovation Council | JSGG20170824152728492 | Mei Guo |

The funders had no role in study design, data collection, and interpretation, or the decision to submit the work for publication.

### Author contributions

Yeming Xie, Data curation, Software, Formal analysis, Validation, Investigation, Visualization, Methodology, Writing – original draft, Writing – review and editing; Fengying Ruan, Resources, Validation, Methodology; Yaning Li, Resources, Validation, Investigation, Methodology; Meng Luo, Validation, Methodology; Chen Zhang, Software, Investigation; Zhichao Chen, Zhe Xie, Weitian Chen, Wenfang Chen, Yitong Fang, Yuxin Sun, Mei Guo, Juan Wang, Investigation; Zhe Weng, Conceptualization, Investigation, Writing – original draft; Shouping Xu, Resources; Hongqi Wang, Resources, Supervision, Funding acquisition, Project administration; Chong Tang, Conceptualization, Resources, Data curation, Software, Formal analysis, Supervision, Funding acquisition, Validation, Investigation, Visualization, Methodology, Writing – original draft, Project administration, Writing – review and editing

### Author ORCIDs

Yitong Fang (ID) http://orcid.org/0000-0001-7255-0958
Chong Tang (ID) http://orcid.org/0000-0002-6898-8946

Joint Public Review: https://doi.org/10.7554/eLife.87868.4.sa1
Author Response https://doi.org/10.7554/eLife.87868.4.sa2

## Additional files

### Supplementary files

- Supplementary file 1. Data table for datasets used in this paper.
- Supplementary file 2. Metrics of pore-C and SCA-seq data in this paper.
- MDAR checklist

### Data availability

Sequencing data have been deposited in NCBI BioProject under the accession code PRJNA917827. All custom codes to run the SCA-seq pipeline are available from GitHub https://github.com/genome-tube/SCA-seq, copy archived at *genometube, 2024*.

The following dataset was generated:

| Author(s) | Year | Dataset title | Dataset URL | Database and Identifier |
| --- | --- | --- | --- | --- |
| Xie Y, Ruan F, Li Y, Tang C | 2023 | Spatial chromatin accessibility sequencing resolves next-generation genome architecture | https://www.ncbi.nlm.nih.gov/bioproject/?term=PRJNA917827 | NCBI BioProject, PRJNA917827 |

The following previously published datasets were used:

| Author(s) | Year | Dataset title | Dataset URL | Database and Identifier |
|---|---|---|---|---|
| Ahn J, Davis ES, Uryu H, Daugird TA, Quiroga IY, Li J, Tsai Y, Parker JS, Zheng D, Legant WR, Phanstiel DH, Wang GG | 2021 | A phase separation mechanism underscores development of cancer and aberrant organization of three-dimensional chromatin structure [Hi-C] | https://www.ncbi.nlm.nih.gov/geo/query/acc.cgi?acc=GSE143465 | NCBI Gene Expression Omnibus, GSE143465 |
| Karabacak A | 2018 | Reproducible inference of transcription factor footprints in ATAC-seq and DNase-seq datasets via protocol-specific bias modeling | https://www.ncbi.nlm.nih.gov/geo/query/acc.cgi?acc=GSE108513 | NCBI Gene Expression Omnibus, GSE108513 |
| Crawford G | 2012 | Experiment summary for ENCSR000EJR | https://www.encodeproject.org/experiments/ENCSR000EJR/ | ENCODE, GSM1008573 |
| Weng Z | 2021 | Summary for annotation file set ENCSR135CRI | https://www.encodeproject.org/annotations/ENCSR135CRI/ | ENCODE, ENCSR135CRI |
| Akhtar A | 2017 | DHX9 suppresses spurious RNA processing defects originating from the Alu invasion of the human genome [RNA-Seq] | https://www.ncbi.nlm.nih.gov/geo/query/acc.cgi?acc=GSE85161 | NCBI Gene Expression Omnibus, GSE85161 |

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
