## [Editor Report · eLife assessment]

This paper reports the development of SCA-seq, a new method derived from PORE-C for simultaneously measuring chromatin accessibility, genome 3D and CpG DNA methylation. Most of the conclusions are supported by **convincing** data. SCA-seq has the potential to become a **useful** tool to the scientific communities to interrogate genome structure-function relationships.

---

## [Referee Report · Joint Public Review]

In this work, Xie et al. developed SCA-seq, which is a multiOME mapping method that can obtain chromatin accessibility, methylation, and 3D genome information at the same time. SCA-seq first uses M.CviPI DNA methyltransferase to treat chromatin, then perform proximity ligation followed by long-read sequencing. This method is highly relevant to a few previously reported long read sequencing technologies. Specifically, NanoNome, SMAC-seq, and Fiber-seq have been reported to use m6A or GpC methyltransferase accessibility to map open chromatin, or open chromatin together with CpG methylation; Pore-C and MC-3C have been reported to use long read sequencing to map multiplex chromatin interactions, or together with CpG methylation. Therefore, as a combination of NanoNome/SMAC-seq/Fiber-seq and Pore-C/MC-3C, SCA-seq is one step forward. The authors tested SCA-seq in 293T cells and performed benchmark analyses testing the performance of SCA-seq in generating each data module (open chromatin and 3D genome). The QC metrics appear to be good and I am convinced that this is a valuable addition to the toolsets of multi-OMIC long-read sequencing mapping.

---

## [Author Response]

The following is the authors’ response to the original reviews.

**eLife assessment**
This paper reports the development of SCA-seq, a new method derived from PORE-C for simultaneously measuring chromatin accessibility, genome 3D and CpG DNA methylation. Most of the conclusions are supported by convincing data. SCA-seq has the potential to become a useful tool to the scientific communities to interrogate genome structure-function relationships.
**Public Reviews:**

**Reviewer #1 (Public Review):**
In this work, Xie et al. developed SCA-seq, which is a multiOME mapping method that can obtain chromatin accessibility, methylation, and 3D genome information at the same time. SCA-seq first uses M.CviPI DNA methyltransferase to treat chromatin, then perform proximity ligation followed by long-read sequencing. This method is highly relevant to a few previously reported long read sequencing technologies. Specifically, NanoNome, SMAC-seq, and Fiber-seq have been reported to use m6A or GpC methyltransferase accessibility to map open chromatin, or open chromatin together with CpG methylation; Pore-C and MC-3C have been reported to use long read sequencing to map multiplex chromatin interactions, or together with CpG methylation. Therefore, as a combination of NanoNome/SMAC-seq/Fiber-seq and Pore-C/MC-3C, SCA-seq is one step forward. The authors tested SCA-seq in 293T cells and performed benchmark analyses testing the performance of SCA-seq in generating each data module (open chromatin and 3D genome). The QC metrics appear to be good and I am convinced that this is a valuable addition to the toolsets of multi-OMIC long-read sequencing mapping.The revised manuscript addressed most of my questions except my concern about Fig. S9. This figure is about a theory that a chromatin region can become open due to interaction with other regions, and the author propose a mathematic model to compute such effects. I was concerned about the errors in the model of Fig. S9a, and I was also concerned about the lack of evidence or validation. In their responses, the authors admitted that they cannot provide biological evidence or validations but still chose to keep the figure and the text.The revised Fig. S9a now uses a symmetric genome interaction matrix as I suggested. But Figure S9a still have a lot of problems. Firstly, the diagonal of the matrix in Fig. S9a still has many 0's, which I asked in my previous comments without an answer. The legend mentioned that the contacts were defined as 2, 0 or -2 but the revised Fig. S9a only shows 1,0, or -1 values. Furthermore, Fig. S9b,9c,9d all added a panel of CTCF+/- but there is no explanation in text or figure legend about these newly added panels. Given many unaddressed problems, I would still suggest deleting this figure.In my opinion, this paper does not need Fig. S9 to support its major story. The model in this figure is independent of SCA-seq. I think it should be spinoff as an independent paper if the authors can provide more convincing analysis or experiments. I understand eLife lets authors to decide what to include in their paper. If the authors insist to include Fig. S9, I strongly suggest they should at least provide adequate explanation about all the figure panels. At this point, the Fig. S9 is not solid and clearly have many errors. The readers should ignore this part.

We appreciate the reviewer for raising these concerns regarding Fig. S9. After careful consideration, we have decided to address your concerns by deleting Fig. S9 and the corresponding text from the manuscript. We understand your point that the model presented in Fig. S9 is independent of SCA-seq and may require additional evidence and validation to be presented in a separate paper.

We agree that it is important to maintain the integrity and accuracy of the manuscript, and we appreciate your feedback in helping us make this decision.

**Reviewer #2 (Public Review):**
In this manuscript, Xie et al presented a new method derived from PORE-C, SCA-seq, for simultaneously measuring chromatin accessibility, genome 3D and CpG DNA methylation. SCA-seq provides a useful tool to the scientific communities to interrogate the genome structure-function relationship.The revised manuscript has clarified almost of the concerns raised in the previous round of review, though I still have two minor concerns,1. In fig 2a, there is no number presented in the Venn diagram (although the left panel indeed showed the numbers of the different categories, including the numbers in the right panel would be more straightforward).

We appreciate the reviewer for pointing out the need for clarification in the Venn diagram in Fig 2a. We have added the numbers to Venn diagram.

1. The authors clarified the discrepancy between sfig 7a and sfig 7g. However, the remaining question is, why is there a big difference in the percentage of the cardinality count of concatemers of the different groups between the chr7 and the whole genome?

We apologize for the confusion regarding the difference in the percentage of the cardinality count of concatemers between chr7 and the whole genome in figures S7a and S7g. The difference arises because the chr7 cardinality count only considers the intra-chromosome segments that are adjacent to each other on a SCA-seq concatemer, while the whole genome cardinality count includes both intra-chromosome and inter-chromosome segments.

In the case of a SCA-seq concatemer that contains both intra-chromosome junctions and inter-chromosome junctions, the whole genome cardinality count will be greater than the intra-chromosome cardinality count. This explains the difference in the percentages between chr7 and the whole genome in figures S7a and S7g.

To better clarify the definition of intra-chromosome cardinality, we have added an illustrative graph in figure S7a. In the updated figure S7a, the given exemplary SCA-seq concatemer has a whole genome cardinality of 4 and a chr7 intra-chromosome cardinality of 3.